# Notable Underlying Mechanism for Pancreatic β-Cell Dysfunction and Atherosclerosis: Pleiotropic Roles of Incretin and Insulin Signaling

**DOI:** 10.3390/ijms21249444

**Published:** 2020-12-11

**Authors:** Hideaki Kaneto, Atsushi Obata, Tomohiko Kimura, Masashi Shimoda, Junpei Sanada, Yoshiro Fushimi, Naoto Katakami, Takaaki Matsuoka, Kohei Kaku

**Affiliations:** 1Department of Diabetes, Endocrinology and Metabolism, Kawasaki Medical School, Kurashiki 701-0192, Japan; obata-tky@med.kawasaki-m.ac.jp (A.O.); tomohiko@med.kawasaki-m.ac.jp (T.K.); masashi-s@med.kawasaki-m.ac.jp (M.S.); gengorou@med.kawasaki-m.ac.jp (J.S.); fussy.k0113@med.kawasaki-m.ac.jp (Y.F.); 2Department of Metabolic Medicine, Osaka University Graduate School of Medicine, Osaka 565-0871, Japan; katakami@endmet.med.osaka-u.ac.jp (N.K.); matsuoka@endmet.med.osaka-u.ac.jp (T.M.); 3Department of General Internal Medicine 1, Kawasaki Medical School, Kurashiki 701-0192, Japan; kka@med.kawasaki-m.ac.jp

**Keywords:** pancreatic β-cell dysfunction, atherosclerosis, incretin signaling, insulin signaling

## Abstract

Under healthy conditions, pancreatic β-cells produce and secrete the insulin hormone in response to blood glucose levels. Under diabetic conditions, however, β-cells are compelled to continuously secrete larger amounts of insulin to reduce blood glucose levels, and thereby, the β-cell function is debilitated in the long run. In the diabetic state, expression levels of insulin gene transcription factors and incretin receptors are downregulated, which we think is closely associated with β-cell failure. These data also suggest that it would be better to use incretin-based drugs at an early stage of diabetes when incretin receptor expression is preserved. Indeed, it was shown that incretin-based drugs exerted more protective effects on β-cells at an early stage. Furthermore, it was shown recently that endothelial cell dysfunction was also associated with pancreatic β-cell dysfunction. After ablation of insulin signaling in endothelial cells, the β-cell function and mass were substantially reduced, which was also accompanied by reduced expression of insulin gene transcription factors and incretin receptors in β-cells. On the other hand, it has been drawing much attention that incretin plays a protective role against the development of atherosclerosis. Many basic and clinical data have underscored the importance of incretin in arteries. Furthermore, it was shown recently that incretin receptor expression was downregulated in arteries under diabetic conditions, which likely diminishes the protective effects of incretin against atherosclerosis. Furthermore, a series of large-scale clinical trials (SPAED-A, SPIKE, LEADER, SUSTAIN-6, REWIND, PIONEER trials) have shown that various incretin-related drugs have beneficial effects against atherosclerosis and subsequent cardiovascular events. These data strengthen the hypothesis that incretin plays an important role in the arteries of humans, as well as rodents.

## 1. Molecular Mechanism for Pancreatic β-Cell Dysfunction

### 1.1. MafA and PDX-1 Play a Crucial Role in Pancreatic β-Cells

Under healthy conditions, pancreatic β-cells function to produce and secrete the insulin hormone in response to high glucose concentrations. Under diabetic conditions, however, β-cells are compelled to secrete larger amounts of insulin continuously in order to reduce blood glucose levels. Such succession is some grueling work for β-cells themselves, and thereby, the β-cell function is debilitated in the long run (so called “pancreatic β-cell glucose toxicity”) [1,2,3]. Chronic hyperglycemia reduces insulin biosynthesis and secretion together with reduced expression of insulin gene transcription factors such as MafA and PDX-1. In clinical practice, it is very important to alleviate such β-cell glucose toxicity so that the aggravation of diabetes mellitus is forestalled. In addition, insulin signaling in insulin target tissues (liver, skeletal muscle, and adipose tissue) is weakened by the burden of glucose toxicity, leading to the development of insulin resistance. Such debilitation of the β-cell function and development of insulin resistance lead to further aggravation of type 2 diabetes mellitus (Figure 1).

MafA is a strong transcription factor for the insulin gene [4,5,6,7,8,9,10,11,12,13,14,15,16,17,18,19]. In MafA knockout mice, insulin biosynthesis and secretion are reduced, leading to diabetes mellitus, indicating the importance of MafA in β-cells [8]. The MafA expression level is markedly reduced in a diabetic state [11], but it is preserved after alleviation of glucose toxicity with some anti-diabetic agent [16,17,18,19]. Furthermore, we reported that in β-cell-specific and conditional (tamoxifen-induced) MafA overexpressing transgenic db/db mice, serum insulin levels were higher and blood glucose levels were lower compared to their littermates. In addition, the β-cell mass was preserved in MafA overexpressing db/db mice mainly due to the reduction of apoptotic cell death. We think that these findings clearly underscore the importance of MafA in β-cells and become unequivocal evidence that downregulation of MafA expression is associated with β-cell glucose toxicity (Figure 1). 

PDX-1, another important transcription factor for the *insulin* gene, plays a crucial role in pancreas development and β-cell differentiation and maintenance of mature β-cell function [20,21,22,23,24,25,26,27,28,29,30,31]. In PDX-1 knockout mice, pancreas formation is not observed at all [23]. In mature β-cells, PDX-1 transactivates several genes including *insulin*, *glucokinase*, and *glut2*. However, the expression level of PDX-1 is reduced in a diabetic state [26,29], which we think is associated with β-cell failure observed in diabetes. Indeed, we recently reported that β-cell-specific and conditional (tamoxifen-induced) PDX-1 overexpressing transgenic Akita mice showed lower blood glucose levels and lower HbA_1C_ values compared to their littermates, accompanied by increased insulin secretion after glucose loading [31]. Expression levels of MafA and glucokinase were preserved in PDX-1 overexpressing Akita mice. We think that such new findings suggest that the downregulation of PDX-1 expression found in diabetes undermines insulin biosynthesis and secretion which explains, at least in part, the mechanism for β-cell glucose toxicity (Figure 1).

### 1.2. Incretin Signaling Plays an Important Role in Pancreatic β-Cells

In response to the ingestion of food, glucagon-like peptide-1 (GLP-1) and glucose-dependent insulinotropic polypeptide (GIP) are released from the gastrointestinal tract both of which stimulate insulin secretion. GLP-1 and GIP bind to the GLP-1 and GIP receptor in β-cells, respectively, which increases intracellular cAMP levels. Increased cAMP levels lead to augmenting insulin secretion, reducing β-cell apoptosis, and facilitating β-cell proliferation (Figure 2). Incretin-related drugs such as glucagon-like peptide-1 receptor (GLP-1R) activators or dipeptidyl peptidase-IV (DPP-IV) inhibitors are known to ameliorate glycemic control and reduce the progression of β-cell dysfunction in human subjects, as well as rodent models [32,33,34,35]. Indeed, it was reported that the GLP-1 receptor activator liraglutide preserved pancreatic β-cell function through the reduction of glucose toxicity, as well as through its direct effect on β-cells [32,35]. The DPP-IV inhibitor suppresses the activity of DPP-IV, which is a splitting enzyme of incretin and increases serum levels of GLP-1 and GIP. Both incretins stimulate insulin secretion in a glucose-dependent manner and GLP-1 suppresses glucagon secretion. Indeed, it has been reported that DPP-IV inhibitors preserve pancreatic β-cell function through the reduction of glucose toxicity, as well as through the direct protective effect of incretin [33,34].

### 1.3. Incretin Receptor Expression in β-Cells Is Downregulated under Diabetic Conditions: Incretin-Based Agents Exert More Protective Effects on β-Cells at an Early Stage of Diabetes rather than an Advanced Stage

It is well known that the incretin hormone action is significantly reduced in subjects with type 2 diabetes mellitus. Moreover, it has been reported that expression levels of incretin receptors are decreased in a diabetic state, which is likely involved in the impaired incretin effects and the progression of β-cell failure found in diabetes [16,36,37,38] (Figure 2). In addition, it has been suggested that the decreased expression of transcription factor 7-like 2 (TCF7L2), which is a transcription factor of incretin receptors and plays a crucial role in the maintenance of β-cell function, is involved in the downregulation of the incretin receptor expression in β-cells found in diabetes [39,40,41]. 

It was reported that incretin-related drug liraglutide increased the β-cell function and mass at an early stage of diabetes, but these effects were attenuated at an advanced stage [35]. Only at an early stage, insulin biosynthesis and glucose-stimulated insulin secretion were markedly increased by liraglutide [35]. In addition, only at an early stage, expression levels of various insulin gene transcription factors such as MafA and PDX-1 were upregulated by liraglutide. We think that the increased expression of such factors at an early stage explains the increased insulin biosynthesis and secretion observed at an early stage. It is likely that the recovery of MafA expression is particularly important for the recovery of β-cell function and amelioration of glycemic control, since MafA regulates not only the insulin gene but also various factors related to the glucose-stimulated insulin secretion. In addition, the expression of GLP-1 receptor was reduced at an advanced stage, which we think explains the reason why liraglutide did not exert beneficial effects at an advanced stage compared to an early stage [35]. Taken together, we think that it is very important to start using incretin-based drugs at an early stage of diabetes to enable incretin-based drugs to fully exert their effect for the protection of β-cell function.

### 1.4. Impaired Insulin Signaling in Endothelial Cells Leads to Pancreatic β-Cell Dysfunction

In general, the main insulin target tissues are the liver, skeletal muscle, and adipose tissues, but there are many kinds of tissues and/or cells in which insulin signaling plays some important role. In endothelial cells, binding of insulin to insulin receptor on its cell surface activates insulin receptor substrate (IRS), phosphoinositide 3-kinase (PI3K), and 3-phosphoinositide-dependent protein kinase-1 (PDK1). Such activated insulin signaling leads to augment nitric oxide production in endothelial cells. Indeed, there have been several reports showing the importance of insulin signaling in endothelial cells [42,43,44,45,46]. Since it is known that the endothelial cell dysfunction is observed under diabetic conditions, it is possible that such endothelial dysfunction brings out hypoxia and ischemia in various tissues through insufficiency of nitric oxide production. In addition, it is known that pancreatic islets are particularly vulnerable to various stimuli including hypoxia and ischemia, and thereby it is also possible that endothelial dysfunction leads to exacerbation of pancreatic β-cell function.

Recently, we examined the possible role of PDK1, one of the important molecules in insulin signaling in vascular endothelial cells, in the maintenance of pancreatic β-cell mass and function. As a result, vascular endothelial-specific PDK1 knockout mice presented reduced β-cell mass and impaired β-cell function [47] (Figure 3). These mice also presented reduced blood flow of pancreas and/or islets and hypoxia of β-cells. In these KO mice, the β-cell mass was significantly reduced and the blood vessel region in islets was significantly decreased. In addition, in these KO mice, incretin secretion was augmented after the oral glucose tolerance test but insulin secretion was impaired, suggesting the impairment of incretin sensitivity in islets of KO mice. Insulin, MafA, PDX-1, GLP-1, and GIP receptor expression levels were all significantly decreased in islets of KO mice [47]. The microsphere experiment elucidated the remarkably reduced islet blood flow, and HIF1α and its downstream factor expression levels were significantly increased in islets of KO mice, indicating that islets of KO mice were in a more hypoxic state compared to the control mice. Consequently, ER stress-related gene expression levels were significantly elevated and inflammatory cytokine levels were increased in islets of KO mice [47].

Taken together, ablation of endothelial PDK1 reduces vascularity in islets, and both pancreatic and islet blood flow are decreased, which lead to hypoxia in islets and induction of ER stress and inflammation. Therefore, it is likely that vascular endothelial PDK1 plays an important role in the maintenance of pancreatic β-cell mass and function by maintaining the vascularity of the pancreas and islets and protecting them from hypoxia, hypoxia-related ER stress, and inflammation. These are novel concepts to explain the underlying molecular mechanism for pancreatic β-cell failure and we think that such findings would be useful when we think about future strategies for type 2 diabetes mellitus (Figure 3). 

## 2. Incretin Signaling and Atherosclerosis

### 2.1. Incretin Signaling Plays an Important Role in Arteries

GLP-1 receptor expression is observed not only in the pancreas but also in a variety of tissues including arteries such as endothelial and smooth muscle cells. In endothelial cells, incretin signaling is known to improve the vascular relaxation response via eNOS expression and activity and retard the development of atherosclerosis [48,49]. The increased expression of PAI-1 and VCAM-1 and induced inflammatory cytokines such as TNF-α under diabetic conditions are suppressed by GLP-1 signaling in endothelial cells. Taken together, it is likely that GLP-1 signaling in blood vessels improves the wall disorder induced by various factors including hyperglycemia and various inflammatory cytokines. Moreover, in vascular smooth muscle cells, GLP-1 receptor stimulation is known to prevent the development of atherosclerosis (Figure 4).

In addition, since the GLP-1 receptor is expressed in various cell types, it has not been clearly elucidated for a long period of time how GLP-1 receptor activators can retard the progression of atherosclerosis. Very recently, however, the vasoprotective mechanism of the GLP-1 receptor activator liraglutide was clearly demonstrated at the cellular level after a series of experiments using global GLP-1 receptor knockout mice, endothelial cell-specific GLP-1 knockout mice, and myeloid cell-specific GLP-1 receptor knockout mice. As a result, the liraglutide treatment normalized blood pressure, cardiac hypertrophy, vascular fibrosis, endothelial dysfunction, oxidative stress, and vascular inflammation in an endothelial GLP-1R-dependent manner, was shown [50]. We think that these recent findings are unequivocally corroborative evidence that the endothelial GLP-1 receptor expression is critical for GLP-1 receptor activators to fully exert their effects in arteries. 

Atherosclerosis and cardiovascular diseases are typical diabetic complications, which sometimes fall into serious and lethal situations. Incretin-based therapy using GLP-1 receptor agonists or DPP-IV inhibitors substantially reduce blood glucose levels without hypoglycemia and/or weight gain, which leads to prevent diabetic macroangiopathy. In addition, as described above, GLP-1 has direct protective effects on vascular cells through the GLP-1 receptor. Therefore, it is likely that incretin-based agents exert beneficial effects against the development of atherosclerosis through both the reduction of blood glucose levels and their direct effects on vascular cells via the GLP-1 receptor. We should be aware of these facts and willingly adopt incretin-based therapy in clinical practice, as well.

### 2.2. Incretin Receptor Expression in Arteries Is Downregulated under Diabetic Conditions

The GLP-1 receptor expression in pancreatic β-cells is reduced under diabetic conditions and TCF7L2 is known to function as a transcription factor for the GLP-1 receptor at least in β-cells. We recently showed that the vascular GLP-1 receptor expression was reduced in a diabetic state as reported in pancreatic β-cells [51] (Figure 4). The GLP-1 receptor and TCF7L2 expression levels in endothelial and smooth muscle cells were significantly lower in obese type 2 diabetic mice. Furthermore, siTCF7L2 decreased the TCF7L2 levels and such reduction of the TCF7L2 level resulted in the downregulation of GLP-1 receptor expression in cultured vascular endothelial cells. In addition, when we overexpressed the TCF7L2 level using the TCF7L2 expressing adenovirus, the GLP-1 receptor expression level was substantially augmented [51,52]. Taken together, the GLP-1 receptor expression level was significantly lower under diabetic conditions, which was accompanied by the reduction of the TCF7L2 expression level. These data also indicate that TCF7L2 is a possible regulator of GLP-1 receptor expression in arteries as reported in β-cells. To the best of our knowledge, this is the first report showing this point, and thus we think such findings would be useful when we create more efficient therapeutic strategies for type 2 diabetes mellitus.

### 2.3. Large-Scale Clinical Trials Regarding the Protective Role of Incretin-Based Agents against Atherosclerosis in Subjects with Type 2 Diabetes Mellitus: SPAED-A and SPIKE Trials

Atherosclerosis is often observed in subjects with type 2 diabetes. In order to predict such atherosclerosis, carotid intima-media thickness (IMT) is often used as an index of the progression of atherosclerosis in clinical practice due to its simplicity and reproducibility without being invasive. It has been reported that the increase of IMT is often observed in diabetic subjects and that there is a close relationship between carotid IMT and the prevalence and extent of coronary stenosis in subjects with type 2 diabetes [53,54,55,56]. Interestingly, it has been shown recently that incretin-based agents actually retard the progression of carotid IMT in subjects with type 2 diabetes (Table 1) [57,58,59,60,61].

(1) SPEAD-A trial [57,58]: The aim of this study was to investigate the effects of DPP-IV inhibitor alogliptin on the progression of carotid atherosclerosis using IMT in subjects with type 2 diabetes. This prospective, randomized, and multicenter study included 341 subjects without a history of overt cardiovascular diseases and observed the effects of alogliptin for 2 years. As a result, changes in the mean common and the right and left maximum IMT of the carotid arteries were significantly greater after the alogliptin treatment compared to the conventional treatment.

(2) SPIKE trial [59,60]: The aim of this study was to investigate the effects of DPP-IV inhibitor sitagliptin on the progression of carotid atherosclerosis using IMT in insulin-treated type 2 diabetic subjects. This prospective, randomized, and multicenter study included 282 insulin-treated subjects without a history of overt cardiovascular diseases and observed the effects of sitagliptin for 2 years. As a result, changes in the mean and left maximum IMT of the common carotid arteries were significantly greater after the sitagliptin treatment compared to the conventional treatment.

There were many basic research reports on the importance of incretin on arteries. To the best of our knowledge, however, the above-mentioned two large-scale clinical trials, SPEAD-A and SPIKE trials, were the first large scale clinical reports demonstrating that incretin-based drugs are actually useful to prevent the development of atherosclerosis in clinical practice. These data clearly indicate that incretin-based agents function to retard the progression of carotid atherosclerosis in subjects with type 2 diabetes without a history of overt cardiovascular disease regardless of the use of insulin therapy. Since it is likely that such findings would lead to the prevention of cardiovascular events in the long run, we should bear in mind in clinical practice that incretin-based agents possess anti-atherosclerotic effects, in addition to original anti-diabetic effects. Thereby, we should willingly adopt incretin-based therapy in clinical practice.

### 2.4. Large-Scale Clinical Trials Regarding the Protective Role of Incretin-Based Agents against Cardiovascular Events in Subjects with Type 2 Diabetes Mellitus: LEADER, SUSTAIN-6, REWIND, PIONEER 6 Trials

While atherosclerosis and subsequent cardiovascular events sometimes fall into serious and lethal situations, it has been reported recently that the administration of GLP-1 receptor activator leads to the reduction of cardiovascular events and cardiovascular-related death (Table 1) [57,58,59,60,61,62,63,64], in addition to atherosclerosis as described above. The importance of incretin-based agents in forestalling the progression of atherosclerosis has been reported in succession very recently, and thus the role of the GLP-1 receptor activator in vascular cells has been capturing the spotlight.

(1) LEADER trial [61,62]: This prospective, randomized, and multicenter study included 9340 subjects and allocated to either the once-daily injection liraglutide or the control group. The median follow-up period was 3.8 years. The primary composite outcome was the first occurrence of cardiovascular death, nonfatal myocardial infarction, or nonfatal stroke. As a result, the primary outcome occurred in significantly fewer patients in the liraglutide group (13.0%) compared to the placebo group (14.9%) (hazard ratio (HR): 0.87, 95% confidence interval (CI): 0.78–0.97). Moreover, fewer subjects died from cardiovascular causes in the liraglutide group (4.7%) compared to the placebo (6.0%) (HR: 0.78, CI: 0.66–0.93). The rate of death from any cause was lower in the liraglutide group (8.2%) compared to the placebo (9.6%) (HR: 0.85, CI: 0.74–0.97). The rates of nonfatal myocardial infarction, nonfatal stroke, and hospitalization for heart failure were relatively lower in the liraglutide group. The rate of the first occurrence of death from cardiovascular causes, nonfatal myocardial infarction, and nonfatal stroke were lower in the liraglutide group compared to the placebo. Taken together, the once-daily injection liraglutide is expected to prevent major adverse cardiovascular events.

(2) SUSTAIN-6 trial [63,64]: This study included 3297 subjects with type 2 diabetes, randomly allocated to either the once-weekly semaglutide or the placebo and was followed up for 2 years. The primary composite outcome was the first occurrence of cardiovascular death, nonfatal myocardial infarction, or nonfatal stroke. As a result, the primary outcome occurred in 6.6% subjects in the semaglutide group and in 8.9% subjects in the placebo group (HR: 0.74, CI: 0.58–0.95). HR (CI) in nonfatal myocardial infarction was 0.74 (0.51–1.08), and that in nonfatal stroke was 0.61 (0.38–0.99). In conclusion, the once-weekly injection semaglutide is also expected to prevent major adverse cardiovascular events.

(3) REWIND trial [65,66]: The aim of this study was to assess the effect of the once-weekly injection dulaglutide on major adverse cardiovascular events in subjects with type 2 diabetes with and without previous cardiovascular disease and a wide range of glycemic control. A total of 9901 participants who had either a previous cardiovascular event or cardiovascular risk factors were randomly assigned to receive the dulaglutide or placebo. The primary outcome was the first occurrence of the composite endpoint of cardiovascular death, non-fatal myocardial infarction, or non-fatal stroke. This study included 9901 participants, and a median follow up period was 5.4 years. As a result, HR (CI) in the primary composite outcome was 0.88 (0.79–0.99), and that in the all cause death was 0.90 (0.80–1.01). Taken together, the once-weekly injection dulaglutide is also expected to prevent major adverse cardiovascular events.

(4) PIONEER 6 trial [67,68]: The aim of this study was to assess the effect of the once-weekly oral semaglutide on major adverse cardiovascular events. A total of 3183 patients were randomly assigned to receive the oral semaglutide or placebo. The primary outcome was the first occurrence of a major adverse cardiovascular event (cardiovascular death, nonfatal myocardial infarction, or nonfatal stroke). As a result, the median time in the trial was 1.3 years. The HR in major adverse cardiovascular events was 0.79 (CI: 0.57–1.11). The other HRs were as follows: Death from cardiovascular causes (HR: 0.49, CI: 0.27–0.92) and all-cause death (HR: 0.51, CI: 0.31–0.84). In conclusion, the once-weekly oral semaglutide is also expected to prevent major adverse cardiovascular events.

The above-mentioned four large-scale clinical trials strongly corroborate the hypothesis that incretin-based agents exert a protective role against cardiovascular events and/or cardiovascular-related death in subjects with type 2 diabetes. Therefore, in clinical practice, we should willingly use incretin-based agents in subjects with type 2 diabetes, especially in subjects with a large risk of atherosclerosis and/or cardiovascular events.

## 3. Conclusions

In this review article, we featured notable underlying mechanisms for pancreatic β-cell dysfunction found in diabetes and the development of atherosclerosis, the first stage of diabetic macroangiopathy leading to cardiovascular events, especially focusing on the pleiotropic role of incretin and insulin signaling in various situations.

Our ideas at present on the pancreatic β-cell dysfunction are as follows: First, in a diabetic state, the expression levels of insulin gene transcription factors (MafA and PDX-1) are downregulated, which we think substantially explains the molecular mechanism for β-cell glucose toxicity. In clinical practice, it is very important to alleviate such β-cell glucose toxicity in order to forestall the exacerbation of diabetes. Second, incretin signaling plays very important roles in pancreatic β-cells, but incretin sensitivity in β-cells is weakened under diabetic conditions, at least in part due to the downregulation of GLP-1 receptor expression, which we think may be associated with an aggravation of the β-cell function. The data also suggest that it would be better to use incretin-related drugs at an early stage of diabetes, and indeed incretin-related drugs exert more protective effects on β-cells at an early stage. Third, endothelial cell dysfunction and subsequent hypoxia or ischemia in β-cells are also likely associated with the β-cell dysfunction found in diabetes. Indeed, after ablation of insulin signaling in endothelial cells, the β-cell function and mass are markedly reduced, which is also accompanied by reduced expression of insulin gene transcription factors and incretin receptors.

Our present ideas on atherosclerosis are as follows: First, incretin signaling plays very important anti-atherosclerotic roles in arteries, but incretin sensitivity in arteries is weakened, at least partially due to the downregulation of GLP-1 receptor expression, which we think may lead to the development of atherosclerosis. Second, a series of large-scale clinical trials have shown that various incretin-related drugs have beneficial effects against atherosclerosis and cardiovascular events. These data strengthen the importance of incretin signaling in arteries of humans, as well as rodents. Taken together, incretin signaling plays a crucial role in various tissues, and incretin-related drugs are promising from clinical points of view, as well as a basic research area.

## Figures and Tables

**Figure 1 ijms-21-09444-f001:**
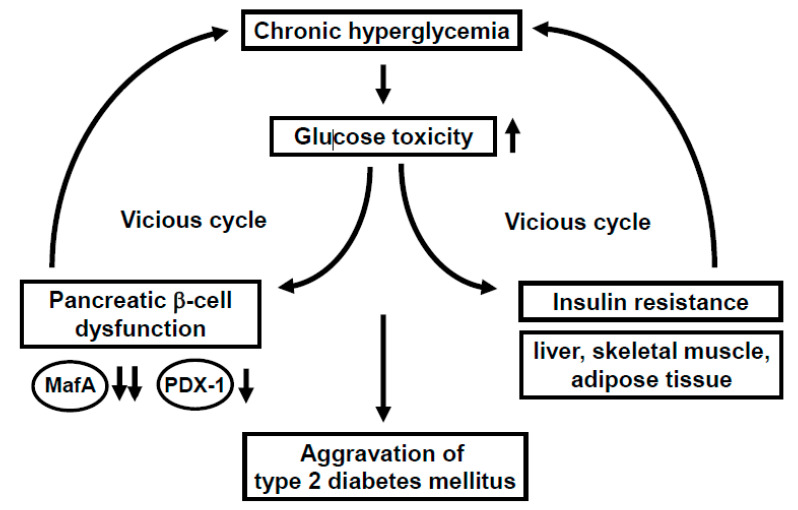
Involvement of glucose toxicity in pancreatic β-cell dysfunction and insulin resistance. Under diabetic conditions, pancreatic β-cell function is gradually debilitated by the burden of glucose toxicity. In addition, insulin signaling in insulin target tissues (liver, skeletal muscle, and adipose tissue) is weakened by the burden of glucose toxicity, leading to the development of insulin resistance. Such debilitation of the β-cell function and development of insulin resistance bring about a vicious cycle and finally lead to further aggravation of type 2 diabetes mellitus.

**Figure 2 ijms-21-09444-f002:**
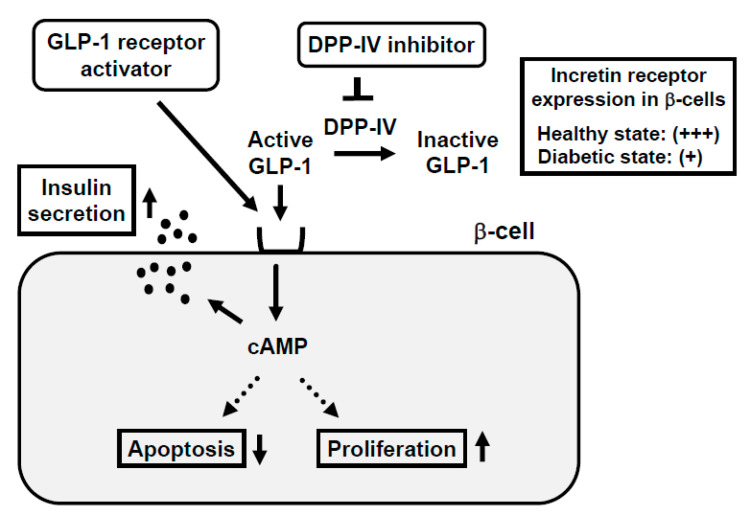
Mechanism of action of incretin and incretin-based agents in pancreatic β-cells and reduced expression of incretin receptor in β-cells in a diabetic state. Binding of incretin to incretin receptor in β-cell membrane increases intracellular cAMP levels, which finally leads to augmentation of insulin secretion from β-cells, reduction of β-cell apoptosis, and facilitation of β-cell proliferation. In a healthy state, the incretin receptor is abundantly expressed in the β-cell membrane, but in a diabetic state, the incretin receptor expression is substantially reduced.

**Figure 3 ijms-21-09444-f003:**
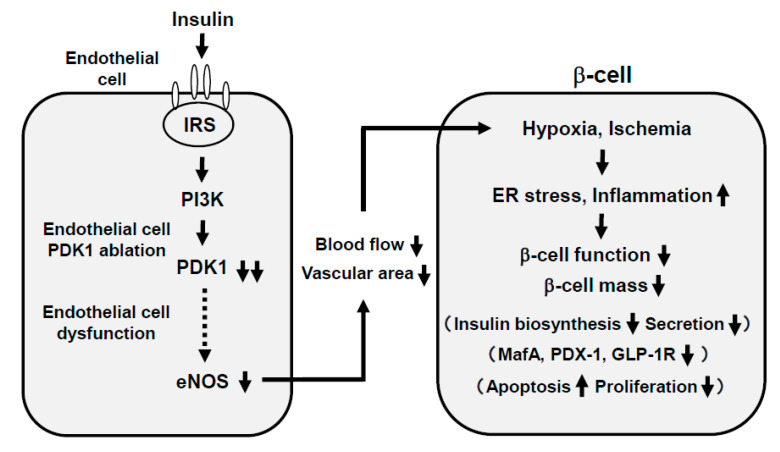
Association of endothelial cell dysfunction with pancreatic β-cell dysfunction: Involvement of hypoxia, provoked endoplasmic reticulum (ER) stress, and inflammatory reaction. Ablation of endothelial PDK1 reduces vascularity in islets, and both pancreatic and islet blood flow are decreased, which lead to hypoxia in islets and induction of ER stress and inflammation. Therefore, it is likely that vascular endothelial PDK1 plays an important role in the maintenance of pancreatic β-cell mass and function by maintaining the vascularity of the pancreas and islets and protecting them from hypoxia, ER stress, and inflammation.

**Figure 4 ijms-21-09444-f004:**
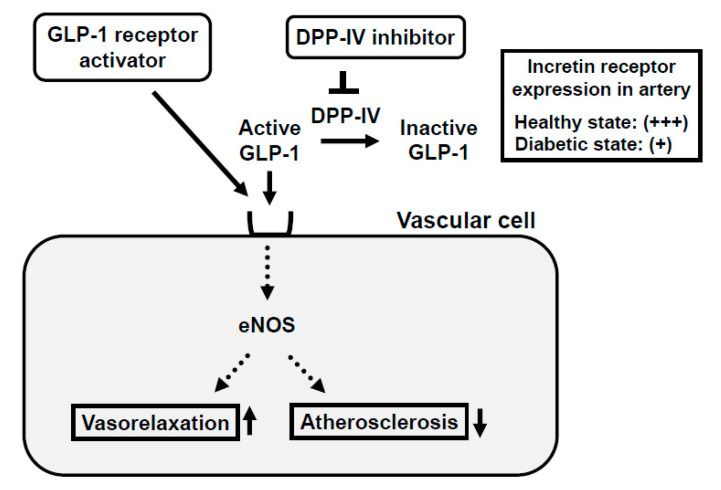
Mechanism of action of incretin and incretin-based agents in vascular cells and reduced expression of incretin receptor in arteries in a diabetic state. Binding of incretin to the incretin receptor in vascular cell membrane facilitates nitric oxide production, which leads to the enhancement of vasodilatory capacity and reduction of atherosclerosis. In a healthy state, the incretin receptor is abundantly expressed in the vascular cell membrane, but in a diabetic state, incretin receptor expression is substantially reduced.

**Table 1 ijms-21-09444-t001:** Outline observation in various large-scale clinical trials with incretin-based agents.

	SPEAD-A	SPIKE	LEADER	SUSTAIN-6	REWIND	PIONEER 6
**Medication**	Alogliptin vs. control	Sitagliptin vs. control	Liraglutide vs. control	Semaglutide vs. control	Dulaglutide vs. control	Oral semaglutide vs. control
**Treatment Period**	2 years	2 years	3.8 years	2 years	5.4 years	1.3 years
**Primary Outcome**	Carotid IMT	Carotid IMT	Cardiovascular outcome	Cardiovascular outcome	Cardiovascular outcome	Cardiovascular outcome
**Results**	Mean IMT *p* = 0.022	Mean IMT *p* = 0.005	Composite outcome HR: 0.87 (CI: 0.78–0.97)	Composite outcome HR: 0.74 (CI: 0.58–0.95)	Composite outcome HR: 0.88 (CI: 0.79–0.99)	Composite outcome HR: 0.79 (CI: 0.57–1.11)
	Right max IMT *p* = 0.025	Left max IMT *p* = 0.021	Cardiovascular death HR: 0.78 (CI: 0.66–0.93)	Nonfatal MI HR: 0.74 (CI: 0.51–1.08)	All-cause death HR: 0.90 (0.80–1.01)	Cardiovascular death HR: 0.49 (CI: 0.27–0.92)
	Left max IMT *p* = 0.013		All-cause death HR: 0.85 (CI: 0.74–0.97)	Nonfatal stroke HR: 0.61 (CI: 0.38–0.99)		All-cause death HR: 0.51 (CI: 0.31–0.84)

IMT: Intima-media thickness; HR: Hazard ratio; CI: Confidence interval; MI: Myocardial infraction.

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
