# Peer review of "Notable Underlying Mechanism for Pancreatic β-Cell Dysfunction and Atherosclerosis: Pleiotropic Roles of Incretin and Insulin Signaling"

_ijms, 2020, doi:10.3390/ijms21249444_

Round 1
Reviewer 1 Report
Hideaki et.al in this review work entitled "Notable underlying mechanism for pancreatic beta-cell dysfunction and atherosclerosis: pleotropic role of insulin and incretin signaling" described about the recent observations in the diabetic and vascular biology with respect to insulin and incretin signaling.
Authors are up to the point and described the importance of incretin in pancreatic and endothelial cell biology and also in the pathological conditions like diabetes and atherosclerosis. Author’s contribution to the reported field is immense and helped the scientific community to further understand the pancreatic biology. With due respect to the authors, here are my comments about the submitted review work.
- 90% of the review is focused on incretin and its role in pancreatic function and atherosclerosis. Inclusion of transcription factors like MafA and PDX-1, and also insulin signaling in pancreatic dysfunction via endothelial cell function looks to little bit distract the readers from the incretin, the major focus of the review. To increase the readability of the review, I suggest authors divide the review in to two major sections, pancreatic dysfunction and endothelial dysfunction/atherosclerosis. Please rearrange the write-up under these two major sections.
- Pancreatic Dysfunction : Sections 1,2,3 and 4
- Atherosclerosis: Sections 5, 6, 7 and 8.
- I suggest authors to prepare two tables for animal and human studies respectively containing outline observations with regard to incretin signaling/role in diabetes and atherosclerosis.
- Though the figures are cleanly depicted, I suggest authors to come up with color figures with possible cell, arteries and organ structures (pancreas).
- Minor typographical errors need to be corrected in the text (using b-cell instead of β-cell).
Reviewer 2 Report
Interesting and well written paper which may add important information about diabetes/atherosclerosis relationship and the possible use of anti-diabetes drugs in chronic heart disease
Author Response
Response to Reviewer 2’s comments
Interesting and well written paper which may add important information about diabetes/atherosclerosis relationship and the possible use of anti-diabetes drugs in chronic heart disease
Thank you very much for your favorable comments.